# The Glycemic Response to Infant Formulas: A Randomized Clinical Trial

**DOI:** 10.3390/nu14051064

**Published:** 2022-03-03

**Authors:** Adi Anafy, Hadar Moran-Lev, Niva Shapira, Meital Priel, Asaf Oren, Laurence Mangel, Dror Mandel, Ronit Lubetzky

**Affiliations:** 1Department of Pediatrics, Dana Dwek Children’s Hospital, Tel Aviv Sourasky Medical Center, Tel Aviv 6423906, Israel; hadarm@tlvmc.gov.il (H.M.-L.); moshe.meital@gmail.com (M.P.); ronitl@tlvmc.gov.il (R.L.); 2Pediatric Gastroenterology Institute, Dana Dwek Children’s Hospital, Tel Aviv Sourasky Medical Center, Tel Aviv 6423906, Israel; 3Sackler Faculty of Medicine, Tel Aviv University, Tel Aviv 6997801, Israel; asafo@tlvmc.gov.il (A.O.); laurencem@tlvmc.gov.il (L.M.); drorm@tlvmc.gov.il (D.M.); 4Department of Nutrition, School of Health Professions, Ashkelon Academic College, Ashkelon 78211, Israel; nivnet@inter.net.il; 5Pediatric Endocrinology and Metabolic Disease Unit, Dana Dwek Children’s Hospital, Tel Aviv Sourasky Medical Center, Tel Aviv 6423906, Israel; 6Department of Neonatology, Dana Dwek Children’s Hospital, Tel Aviv Sourasky Medical Center, Tel Aviv 6423906, Israel

**Keywords:** infant formula, cow’s milk protein-based formula, soy protein-based formula, lactose-free formula, glycemic index

## Abstract

Background: Commercial infant formulas attempt to imitate human milk’s unique composition. However, lactose-free and milk protein-free formulas are often chosen due to medical reasons or personal preferences. The aim of this study was to determine the glycemic and insulinemic indices of a variety of infant formulas. Methods: We conducted a three-arm, randomized, double-blind, crossover study. Participants were 25–40-year-old healthy adults. Three commercial infant formulas (cow’s milk protein-based [“standard”], soy protein-based, and lactose-free) were randomly given to each participant. Glycemic and insulinemic responses were determined and compared between the three formulas. Results: Twenty subjects were enrolled (11 females/9 males, mean age 32.8 ± 2.9 years). No significant difference was found in the glycemic index between the three formulas (21.5, 29.1, and 21.5 for the standard, soy protein-based, and lactose-free formulas, respectively, *p* = 0.21). However, maximal glucose levels were significantly higher for the soy protein-based formula compared to both the standard and lactose-free formulas (111.5 compared to 101.8 and 105.8 mg/dL, respectively, *p* = 0.001). Conclusion: Cow’s milk protein-based, soy protein-based, and lactose-free formulas have a similar glycemic index. However, soy protein-based formula produced a significantly higher increase in postprandial glucose levels. The implication and biological significance of these results have yet to be determined.

## 1. Introduction

Human milk (HM) is recommended as the sole source of nutrition for term infants up to the age of 6 months and supplemented with solid foods for up to two years or beyond [1,2,3]. Commercial infant formulas try to imitate HM unique composition. After water, carbohydrates are the most prevalent nutrient in human milk, in the form of lactose and oligosaccharides [4]. The two types of HM proteins, whey and casein, differ in their solubility in an acidic environment [5].

Both lactose and milk proteins contribute to HM’s low glycemic index (GI) [6,7]. Low-glycemic-index foods are a desirable component of nutrition in the modern era, and epidemiological evidence suggests many benefits of a low-glycemic-index diet [8,9,10]. However, both lactose and milk proteins are often replaced in infant formulas by substitutes due to medical reasons, such as metabolic diseases (e.g., galactosemia) and cow’s milk protein allergy [11,12], as well as due to personal preferences (e.g., religion and veganism). The main alternatives of carbohydrates in infant formulas are sucrose, corn syrup, and tapioca or corn starch. One of the alternatives to milk protein is a soy protein-based formula.

There is little information on the metabolic response to breast milk consumption, and on whether it differs among various formulas. To the best of our knowledge, only one study [13] demonstrated similar glycemic and insulinemic responses in adults who consumed breast milk and cow’s milk-based (“standard”) formula, with a wide range of responses to other formulas in a different cohort. 

The aim of the present study was to investigate the glycemic and insulinemic responses to several common commercially available types of formulas, containing carbohydrate or protein substitutes, in comparison to those of a standard formula based on cow’s milk protein and lactose. We hypothesized that the glycemic and insulinemic responses to a soy protein-based formula and a lactose-free formula will be different from those following the ingestion of the standard formula.

## 2. Materials and Methods

### 2.1. Patient Population

This study was conducted from March 2018 to December 2019 at the Tel Aviv Sourasky Medical Center. Due to ethical considerations and practical reasons (e.g., consuming different types of infant formulas in defined quantities and taking repeated blood samples), this study could not be conducted on infants. Therefore, the participants were healthy young adult volunteers aged 25–40 years, with a normal body mass index (BMI). Candidates were excluded if they had chronic illnesses or metabolic morbidities (e.g., diabetes, insulin resistance, or abnormal fasting glucose values), if they were overweight (BMI > 25 kg/m^2^), if they had taken any medication on a regular basis, and if they had a soy allergy or lactose sensitivity or avoided dairy products for any reason (e.g., veganism). Pregnant women were also excluded. Each subject served as his/her own control.

### 2.2. Study Design

This study is a three-arm, double-blind, randomized, crossover trial. Upon study inclusion, anthropometric (weight, height, and BMI) and demographic data (age and sex) were recorded for each participant. The study design followed the standardized method for the determination of the glycemic index of foods [14]. Each participant completed a total of four sessions: 50 g of glucose dissolved in water (reference food) was consumed in the first session, and three different kinds of formulas were randomly consumed in the next three sessions. The three formulas included a “standard” formula, which was a cow’s milk protein-based formula (“Nestlé, Materna Extra Care Stage 1”) containing lactose, a soy protein-based formula (“Nestlé, Materna Soya”), and a lactose-free formula (“Nestlé, Materna Extra Care Comfort”). All formulas contained 50 g of carbohydrates and 9 g of proteins for a total volume of 200 mL each. Table 1 depicts the formulas’ carbohydrate and protein compositions.

At each of the sessions, each participant had an intravenous line placed, and underwent a blood test for fasting glucose and insulin (at time 0). In the first session, blood samples for liver enzymes, lipid profile, and HbA1C were taken in order to rule out unknown morbidity that could affect glycemic or insulinemic responses. Each participant then consumed 200 mL of concentrated glucose solution containing 50 g of glucose, over a period of up to two minutes. Blood glucose and insulin levels were measured at defined time points over a two-hour period: glucose levels were taken at seven time points (fasting (baseline) and 15, 30, 45, 60, 90, and 120 min after drinking the solution), and insulin levels at three time points (fasting (baseline) and 60 and 120 min after drinking the solution). These glucose and insulin values were taken as references of the glycemic and insulinemic responses for each participant, as defined elsewhere [14].

After a washout period, each volunteer participated in three additional sessions—in each session, he or she was randomly assigned one of the three formulas above. An independent medical team member was responsible for randomization (performed by a computer software) and formula preparation (prepared by mixing powder into water), so that both the participant and the researcher providing the formulas and taking the blood samples were blinded to the formula the participant was receiving at each session. There was a 48 h or longer break between sessions, and each trial was conducted after a 10 h night fast. Blood samples were taken as described above in each of these three sessions. The participants neither ate nor drank and remained seated during each session, which lasted approximately 120 min.

### 2.3. Statistical Analyses

The SPSS software was used for all statistical analysis (IBM SPSS statistic for Windows, version 25, IBM corp., Armonk, NY, USA, 2017). Categorical variables were reported as frequency and percentage. Normality was assessed by the Kolmogorov–Smirnov test and with a histogram. Continuous variables are presented as the medians and interquartile ranges (IQR) or the mean ± standard deviation (SD) as appropriate. Chi-squared tests or Fisher’s exact test were used for categorical variables as appropriate. GI variations between the formulas were evaluated by the Kruskal–Wallis test. Because each of the tested formulas was given to each participant, the Friedman test and the Wilcoxon test were applied to compare postprandial glucose and insulin levels between the different solutions.

The glycemic index was calculated using the area under the curve over the baseline, excluding the area beneath the baseline (incremental area under the curve [IAUC]), as recommended by the Food and Agriculture Organization, which was assessed as the sum of the trapezium, relative to the IAUC after reference food consumption (glucose solution) [14]. All statistical tests were two tailed, and a *p* value < 0.05 was considered significant.

### 2.4. Ethical Considerations

The study protocol was approved by the institutional review board of the Tel Aviv Sourasky Medical Center (0760-16-TLV) and by the Israel Ministry of Health (MOH_2017-07-13_000631, https://my.health.gov.il/CliniTrials/Pages/MOH_2017-07-13_000631.aspx (accessed on 28 February 2022)). Written informed consent was obtained from all participants.

## 3. Results

A total of 31 volunteers were originally recruited to this study. Eleven of them were excluded because they failed to meet the inclusion criteria. The study cohort consisted of 20 healthy adults. Nine of them were males (45%) and 11 were females (55%), their mean ± SD age was 32.8 ± 2.9 years, and their median (IQR) BMI was 21.1 (19.7–23.4). Their lipid profile and liver function test results were all within normal limits, and their median HbA1C was 5.3% (5.1–5.6) (normal range 4.6–5.7%) (Table 2).

The participants completed the four test sessions with no adverse events. Figure 1 and Figure 2 depict the 2 h glucose and insulin response curves for the three infant formulas. There was no significant difference in the calculated glycemic index between the three formulas (21.5 ± 21.7, 29.1 ± 17.2, and 21.5 ± 14.5 for the standard formula, soy protein-based formula, and lactose-free formula, respectively, *p* = 0.21, Figure 3 and Table 3). However, the calculated mean of peak glucose levels reached by the participants during the two-hour period was significantly higher for the soy protein-based formula than for the cow’s milk protein-based or lactose-free formulas (111.5 vs. 101.8 and 105.8 mg/dL, respectively, *p* = 0.001, Table 3). Moreover, the maximum increase in glucose levels after formula consumption relative to baseline was also significantly higher for the soy protein-based formula than for the standard or lactose-free formulas (21.7 vs. 13.1 and 16.3 mg/dL, respectively, *p* = 0.006, Table 3). No significant difference was found for the minimum glucose levels between the formulas (78.7 mg/dL, 75.1 mg/dL, and 79.2 mg/dL for the standard formula, soy protein-based formula, and lactose-free formula, respectively, *p* = 0.47, Figure 1).

The postprandial insulinemic response, during the two-hour period, was similar for all three formulas used. The maximum insulin levels and the maximum insulin increase from baseline for the standard formula, soy protein-based formula, and lactose-free formula were 12.1, 12.1, and 10.8 mU/mL (*p* = 0.45, Figure 2), and 5.0, 5.0, and 3.4 mU/mL (*p* = 0.52, Table 3), respectively.

## 4. Discussion

In this randomized, double-blind controlled study, we aimed to assess the glycemic and insulinemic responses to three main types of formula. We found no differences in the glycemic index between the standard (cow’s milk protein-based containing lactose), the lactose-free, and the soy protein-based formula. However, peak blood glucose levels were significantly higher after the consumption of a soy protein-based formula.

It is agreed upon that if certain stimuli occur during the sensitive and critical period of the first 1000 days from the time of fertilization to the age of two years, they can lead to adaptive changes, known as “metabolic programming” [15,16]. These changes have been shown to influence obesity and metabolic morbidity later in life [17,18]. This highlights the importance of assessing the specific glycemic response that is elicited by consuming various types of infant formula during this critical period.

Epidemiological evidence suggests that a diet based on low-glycemic-index carbohydrates has many benefits, especially in an era when obesity is becoming a global epidemic [8,9,19]. Several studies have examined and demonstrated the effect of certain carbohydrates and amino acids on postprandial glycemia and insulinemia [6,20,21,22,23,24,25], but there is little information on the differences in metabolic response to various infant formulas. Wright et al. [13] compared glycemic and insulinemic responses to breast milk and standard infant formula among 10 healthy breastfeeding mothers, and found no differences between them. In that part of their study, similarly to ours, each volunteer consumed both her own breast milk and infant formula. In the second part of their study, 11 formulas that differed in carbohydrate and protein composition were tested among 10 healthy young adult volunteers, and a wide range of responses was demonstrated. It is well known that the metabolic response differs among people, and it is possible that this also affected the differences observed by Wright et al. Our current double-blind study, which included almost twice as many volunteers, compared the glycemic and insulinemic responses to different kinds of infant formula, but did so while eliminating the variability in metabolic responses between different individuals, since each subject served as his/her own control.

To the best of our knowledge, our study is the first to compare glycemia and insulinemia as the result of consuming standard formula with formulas containing carbohydrate or protein substitutes in healthy adults. Our findings might be clinically relevant because high postprandial glucose levels, especially during the growth period and metabolic programming in infancy, might have long-term adverse effects. The present study showed that standard cow’s milk formula, lactose-free formula, and soy protein-based formula had a similar glycemic index despite their different compositions. However, our findings demonstrated that the consumption of a soy protein-based formula resulted in a significantly higher increase in postprandial peak glucose levels. The implications of similar glycemic indexes but different peak glucose levels have yet to be explored, but they may indicate rapid changes in blood glucose concentrations following the consumption of a soy protein-based formula. These fluctuations in glucose levels may increase hunger levels, cause rebound hypoglycemia, and impair the body’s ability to maintain stable blood glucose levels over time [19]. Noteworthy, although postprandial peak glucose levels were significantly higher for the soy-based formula, it did not meet the criteria for hyperglycemia.

Despite the increase in peak glucose levels following the consumption of a soy protein-based formula, the maximum increase in insulin levels was similar for all tested formulas. The fact that we only had three insulin measurements throughout the two-hour period in each session, as opposed to seven glucose tests, may have reduced accuracy in the assessment of the insulin response. 

Lactose is the principal carbohydrate in breast milk, and its many benefits are well known. For example, it enhances calcium absorption (thus contributing to bone mineralization) as well as the absorption of other minerals, such as magnesium, zinc, and iron [26,27,28]. In addition, some of its derivatives and fermentation process products have a prebiotic potential, and are utilized by the gut bacteria from probiotic strains thus encouraging their growth. Furthermore, galactose, a monosaccharide that is one component of lactose, plays an important role in the brain and nervous system development, and is a vital source of energy for the brain, especially in neonates [29,30,31]. Therefore, regardless of its glycemic index, the removal of lactose from the infant’s diet should be carefully considered if done without a compelling medical cause.

The soy protein-based formula is medically recommended for infants suffering from galactosemia (a rare hereditary metabolic disorder) or lactase deficiency (hereditary or acquired). However, it is otherwise not recommended for use during the first 6 months of life [32]. Soy is recognized as being one of the richest sources of phytoestrogens, which are plant ingredients similar in composition to mammalian estrogens [33,34]. In addition, soy products are relatively rich in phytate, which binds calcium and impairs its absorption, and a soy-based formula contains significantly more aluminum than other formulas [35,36]. Our present study may rise another reason to refrain from using a soy protein-based formula without justified medical indication.

The main limitation of our study is that the participants were adults and not infants who are the formula consumers and the research target. Unfortunately, this is unavoidable due to obvious ethical and practical reasons. We are well aware of the differences between the metabolism of infants and that of adults and the age-related differences in the enzymatic activity in the digestive system, which can affect the glycemic and insulinemic responses. However, it is reasonable to assume that infants will respond to these various formulas in a similar manner, if not in a more pronounced way. Another limitation lies in the assessment of the insulinemic response; its accuracy may have been hampered by budgetary constraint, because only three insulin tests per participant in each session were performed. A major strength of this study is that each subject served as his/her own control for the evaluation of each formula.

## 5. Conclusions

In conclusion, the hypothesis that formulas containing carbohydrate or protein substitutes would have different glycemic indexes than a typical formula was not supported by this study. Instead, this study revealed a significant increase in blood glucose levels after the consumption of soy protein-based formula when compared to standard and lactose-free formulas. The implication and clinical significance of these results have yet to be determined; however, we believe that these findings are of concern in the decision-making process for pediatricians considering endorsing a soy protein-based formula without specific medical justification.

## Figures and Tables

**Figure 1 nutrients-14-01064-f001:**
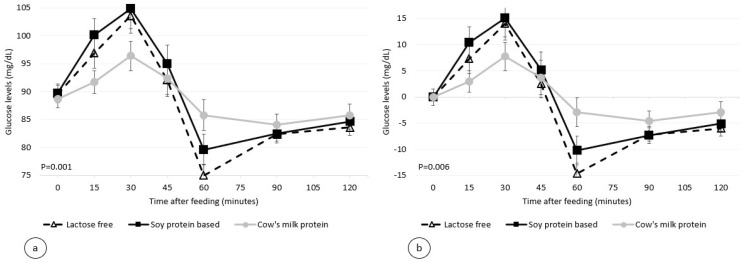
Postprandial plasma glucose level per infant formula (mean ± SEM (standard error of the mean)), (**a**) absolute values, (**b**) relative to the baseline.

**Figure 2 nutrients-14-01064-f002:**
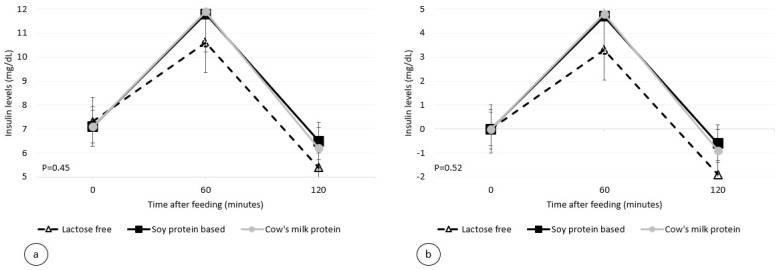
Postprandial plasma insulin level per infant formula (mean ± SEM): (**a**) absolute values; (**b**) relative to the baseline.

**Figure 3 nutrients-14-01064-f003:**
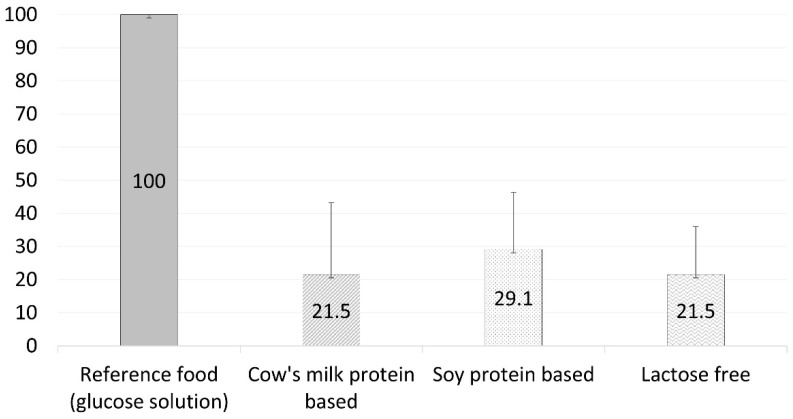
Glycemic index per infant formula (mean ± SD).

**Table 1 nutrients-14-01064-t001:** Carbohydrate and protein content of infant formulas.

Formula Brand	Carbohydrate Composition *	Protein Composition *
Cow’s milk protein-based formula (Materna Extra Care Stage 1)	Lactose (100%)	Whey (60%), casein (40%)
Soy protein-based formula (Materna Soya)	Glucose syrup solids (100%)	Soy (100%)
Lactose-free formula (Materna Extra Care Comfort)	Glucose syrup solids (100%)	Whey (60%), casein (40%)

* Values taken from nutritional information on product labels.

**Table 2 nutrients-14-01064-t002:** Demographic and clinical characteristics.

		Participants (n = 20)
Sex		
	Male	11 (55%)
	Female	9 (45%)
Age, year (range)		32.8 ± 2.9 (28–38)
Anthropometric data		
	Height (m)	1.7 (1.6–1.8)
	Weight (kg)	61.5 (58.0–73.8)
	BMI (range)	21.1 (19.7–23.4)
Laboratory data		
	AST (U/L)	21.5 (18.0–25.8)
	ALT (U/L)	21.0 (13.3–25.0)
	TG (mg/dL)	74.0 (55.8–91.8)
	Total cholesterol (mg/dL)	172.0 (158.3–193.5)
	HDL (mg/dL)	56.6 (44.9–64.0)
	LDL (mg/dL)	103.0 (81.3–113.3)
	HbA1C (%)	5.3 (5.1–5.6)

Data are presented as n (%), the mean ± SD, or the median (Q1–Q3). BMI—body mass index, AST—aspartate transaminase, ALT—alanine transaminase, TG—triglycerides, HDL—high-density lipoprotein, LDL—low-density lipoprotein, and HbA1C—hemoglobin A1C.

**Table 3 nutrients-14-01064-t003:** Postprandial differences in glucose and insulin levels per formula.

	Cow’s Milk Protein Formula	Soy Protein-Based Formula	Lactose-Free Formula	*p* Value
Glycemic index	21.5 ± 21.7	29.1 ± 17.2	21.5 ± 14.5	0.21
Peak glucose level (mg/dL)	101.8 ± 9.1	111.5 ± 13.7	105.8 ± 12.1	0.001 *
Glucose changes from baseline (mg/dL)	13.1 ± 7.2	21.7 ± 11.5	16.3 ± 10.3	0.006 *
Peak insulin level (mU/mL)	12.1 ± 6.5	12.1 ± 7.0	10.8 ± 5.5	0.45
Insulin change from baseline (mU/mL)	5.0 ± 5.7	5.0 ± 5.2	3.4 ± 6.4	0.52

Values are expressed as the mean ± standard deviation. * Difference between soy-based formula and both cow’s milk protein-based and lactose-free formulas.

## Data Availability

The datasets used and/or analyzed during the current study are available from the corresponding author on reasonable request.

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
