# Peer review of "The Glycemic Response to Infant Formulas: A Randomized Clinical Trial"

_nutrients, 2022, doi:10.3390/nu14051064_

Round 1

Reviewer 1 Report

This is an interesting study looking at the GI and glucose / insulin responses to consumption of different infant formula. It is very well written  - clear and concise. It is informative and recognises the limitations and scope of the results. I just have a few minor suggestion for improving the manuscript.

Abstract - line 31 - rephrase 'elicited a similar glycemic index'. This sounds strange as we can elicit a glucose response but not a GI. A GI is attributed to the food not the response.

Introduction - line 54 - remove 'a' before 'similar'

Methods - line 72. BMI needs units (kg/m2) and 'base' should be 'basis'

-Line 77 - this is the first time a pilot study is mentioned. On the basis of these results is there need to carry out a larger study? If not, I would remove reference to 'pilot study'. A small study is fine if that is all that is needed.

-How fast were the glucose and milk drinks consumed and was this standardised? This should be stated.

  • Line 105 - he/she

Results

 - line 158 -I am interested in the results relative to baseline. Is it possible to present a figure showing the changes in glucose concentration relative to baseline. This might show the differences between milks better than the actual glucose concentrations (Figure 1).

Discussion

Line 243 - In the UK soya formula is rarely medically recommended and hydrolysed formula is recommended for cow milk protein allergy. Soya milks are not recommended as an alternative for infants under 6 months . So it might be worth stating some recommendations from different countries and whether they agree.

Author Response

We wish to thank the reviewer for these kind words. Our thanks, too, for the in-depth analysis of our work and for raising several important points that require clarification. We appreciate the time and effort expended. We addressed each issue that was raised as follows:

  1. Abstract - line 31 - rephrase 'elicited a similar glycemic index'. This sounds strange as we can elicit a glucose response but not a GI. A GI is attributed to the food not the response.

The sentence was corrected and now reads “have a similar glycemic index”, line 31.

  1. Introduction - line 54 - remove 'a' before 'similar'.

The sentence was corrected and ‘a’ was removed, line 54

  1. Methods

Line 72 - BMI needs units (kg/m2) and 'base' should be 'basis'.

The units (kg/m2) were added, and the word 'base' was changed to 'basis', line 72.

Line 77 - this is the first time a pilot study is mentioned. On the basis of these results is there need to carry out a larger study? If not, I would remove reference to 'pilot study'. A small study is fine if that is all that is needed.

The reviewer's point is very well taken. The word 'pilot' has been removed, line 78.

How fast were the glucose and milk drinks consumed and was this standardised? This should be stated.

The reviewer’s point is very well taken. We now added: "over a period of up to two minutes", lines 98-99.

Line 105 - he/she

The sentence was corrected and now reads “he or she”, line 106.

  1. Results - line 158 -I am interested in the results relative to baseline. Is it possible to present a figure showing the changes in glucose concentration relative to baseline. This might show the differences between milks better than the actual glucose concentrations (Figure 1).

Many thanks to the reviewer for providing such a thorough analysis of our work. In response to your comments, we have added a figure that illustrates the change in glucose values compared to the baseline (Figure 1b). The same was done regarding the change in insulin level according to the baseline value (Figure 2b).

  1. Discussion - line 243 - In the UK soya formula is rarely medically recommended and hydrolysed formula is recommended for cow milk protein allergy. Soya milks are not recommended as an alternative for infants under 6 months . So it might be worth stating some recommendations from different countries and whether they agree.

Thank you for this valuable and informative comment. In Israel, as in the UK and in many other countries, soy protein-based formulas are seldom used and are not recommended for children under 6 months for the same reasons outlined in lines 245-252. Although, as described in the introduction (lines 48-49), some people use this formula based on personal preferences (religion, veganism and the belief that it would alleviate colic). Obviously, the purpose of our study is not to advocate the consumption of soy protein-based formulas, but rather to add another consideration that should be taken into account before using it for anything other than purely medical use.

Reviewer 2 Report

A contribution to preventing obesity by optimising infant nutrition is important. The paper emphasises that breastmilk is optimal, but examines the glycemic index of various infant formula that are required if breastmilk is unavailable.

Using participants as their own control adds strength to the conclusions. I would have liked to have seen breastmilk included as well as the different kinds of formulae to confirm the results of Wright et al (ref 13). The authors do include in the limitations that the participants were adult and not infants, understandably due to ethical considerations.

Minor corrections:

Page 3 paragraph 2 “he was” should be “he or she was”

Table 2 Age, year is given only as mean ± SD not range

Author Response

We wish to thank the reviewer for drawing our attention to several points in our work that need clarification. We dealt with each one as follows:

  1. Page 3 paragraph 2 “he was” should be “he or she was”

The sentence was corrected and now reads “he or she”, line 106.

  1. Table 2 Age, year is given only as mean ± SD not range

The range was added (Table 2, page 4).